# The economic burden of loiasis: A comprehensive cost-of-illness analysis of regionally representative, individual-level data from rural Gabon

Cédric Isaac Mbavu[1,2,3,4], Kerstin Perlik[1,5], Tom Stargardt[4], Selidji Todagbe Agnandji[6], Olouyomi Scherif Adegnika[6,7], Rella Zoleko-Manego[2,3,6,8], Michael Ramharter[2,3,6,8], Jan Priebe[1,4]*

**1** Research Group Health Economics, Bernhard Nocht Institute for Tropical Medicine (BNITM), Hamburg, Germany, **2** Clinical Research Department, Bernhard Nocht Institute for Tropical Medicine (BNITM), Hamburg, Germany, **3** I. Department of Medicine, University Medical Center Hamburg-Eppendorf (UKE), Hamburg, Germany, **4** Hamburg Center for Health Economics (HCHE), University of Hamburg, Hamburg, Germany, **5** Faculty of Economics and Management, Free University of Bolzano-Bozen, Bozen, South Tyrol, Italy, **6** Centre de Recherches Médicales de Lambaréné (CERMEL), Lambaréné, Gabon, **7** Fondation pour la Recherche Scientifique (FORS), Cotonou, Benin, **8** German Center for Infection Research (DZIF), Partner Site Hamburg-Lübeck-Borstel-Riems, Hamburg, Germany

\* jan.priebe@bnitm.de

## Abstract

Loiasis is a vector-borne filarial infection endemic to parts of sub-Saharan Africa. It disproportionately affects economically disadvantaged communities in rural, forested regions. To better understand the economic burden of loiasis, we conducted a comprehensive cost-of-illness study in an endemic region of Gabon, with the aim of quantifying the financial costs incurred by individuals infected with the disease from a societal perspective. We conducted a cross-sectional survey in 2023 in rural Gabon. Study participants took part in diagnostic testing for loiasis and were interviewed based on a standardized questionnaire covering a wide range of medical and non-medical costs. Participants reporting eye worm migration or harboring loiasis microfilariae were defined as loiasis positive. Various cost estimates were derived by creating a synthetic control group by means of entropy-balancing and then applying generalized linear models (GLM) for the study region. We show that the average annual costs directly attributable to loiasis amount to 39.94 USD per individual per year. Average cost estimates are primarily driven by indirect costs and direct non-medical costs. We further show that in the rarer cases that individuals seek treatment at formal or informal healthcare providers for loiasis-specific symptoms, costs from the patient's perspective can be excessively high and amount to about 43 percent of the average monthly per capita income in the study region.

**Data availability statement:** All data and programming files will be made available from the GESIS repository for the social sciences.

**Funding:** Funding to support this research was provided by the Bernhard Nocht Institute for Tropical Medicine (BNITM). Recipients of this funding were Michael Ramharter and Jan Priebe. Additional financial support was provided by the German Center for Infection Research (FKZ: 8008803909). Recipient of this grant was Michael Ramharter.

**Competing interests:** The authors have declared that no competing interests exist.

## 1. Introduction

Loiasis, a parasitic infection transmitted by deerflies, is estimated to affect more than 20 million individuals in the rural parts of West and Central Africa [1,2]. Initially considered a benign condition whose main policy relevance lay in complicating treatment for onchocerciasis (river blindness) and lymphatic filariasis in cases of co-infection [3,4], loiasis has received limited attention. However, recent studies demonstrate that the infection can have severe health consequences, including a substantial burden of disease [4], impaired cognition [5], organ failure [6], and even excess mortality [7,8].

The high number of infected individuals in Africa, combined with growing evidence of the adverse health effects of loiasis, has led many scholars to call for its inclusion on the WHO's list of neglected tropical diseases (NTDs) [1,8–10]. This demand is further reinforced by recent epidemiological data showing the infection's increasing presence and repeatedly documented negative consequences.

Few studies have yet attempted to estimate the economic burden of loiasis and shed light on the direct and indirect costs attributable to the disease. While several studies report that loiasis patients suffering from acute symptoms (e.g., migration of the adult worm through the conjunctiva of the eye, Calabar swelling or headaches [11,12]) seek treatment from health professionals including nurses, doctors, and traditional healers [13,14], cost estimates are not readily available. Some preliminary evidence on costs is provided by Veletzky and colleagues. Focusing exclusively on loiasis-positive individuals in an endemic region of rural Gabon and obtaining item-specific price data from small sub-samples, the authors simulate loiasis-specific treatment costs under various scenarios. According to the authors' preferred specification, loiasis resulted in treatment costs of 58 United States Dollars (USD) per person per year [14].

Conducting a comprehensive and methodologically sound cost-of-illness analysis based on novel primary data, the aim of this study is to provide a full, empirical estimate of the incremental direct and indirect costs attributable to loiasis in an endemic region of Gabon from a societal perspective. Based on the collected data, estimates for direct medical costs, direct non-medical costs, and indirect costs related to absenteeism and presenteeism are generated. These cost estimates rely on the human capital approach, which considers the entire period during which an individual is absent due to the disease when calculating the economic burden [15]. This approach is implemented by using a two-step statistical procedure involving i) a synthetic control group generated by entropy balancing [16] and ii) generalized linear models (GLM) regression techniques, which are considered the standard method for analyzing healthcare costs [17].

Our findings suggest that loiasis is associated with sizeable direct non-medical and indirect costs in economically disadvantaged communities. Notably, patients with severe symptoms can incur treatment costs that are often catastrophic, underscoring the need for a reevaluation of loiasis as a significant public health concern. These results necessitate a more concerted research effort and policy measures to mitigate the impact of loiasis on affected populations.

## 2. Methods

### 2.1. Data sources

For this cost-of-illness study, we implemented a cross-sectional survey that was conducted from 19 April 2023–11 July 2023 in the three Gabonese provinces of Estuaire, Moyen-Ogooué and Ngounié. Previous studies described the region's endemicity of loiasis and *Mansonella perstans* [18,19] and highlighted that transmission of lymphatic filariasis is low or perhaps absent [20]. Written informed consent was obtained from each participant or their legal representative and tablet-based interviews were conducted by local enumerators that were extensively trained prior to the start of the survey.

Adapting established cost-of-illness survey modules to the local Gabonese context, the survey comprised an extensive set of questions in which respondents were asked to report on (i) health condition, including various infectious and chronic diseases, (ii), healthcare seeking behavior, including formal and informal healthcare supplies such as traditional healers, (iii) incurred direct and indirect financial and in-kind expenses on inpatient and outpatient medical treatment, (iv) jobs and labor market participation, and (v) overall earnings, incomes, and expenditures. As detailed in S13 Text, we complemented self-reported data on health expenditures with inpatient and outpatient costs covered by the national public health insurance (CNAMGS or Caisse nationale d'assurance maladie et de garantie sociale) in order to cover the full cost that a given medical treatment implied. All monetary values are converted into USD using the official exchange rate from 11 July 2023 (1 USD = 597.762 CFA) [21].

Biological material was collected from study participants directly after the completion of the survey. Following established practices, a trained study nurse collected venous blood from the voluntary participants between 10:00 am and 03:00 pm [4]. Two thick blood smears were then prepared for each of those participants with 50 µl of blood. The slides were processed and stained with distilled water, methanol and 10% Giemsa, respectively. After the slides were dry, one of them was read by a trained microscopist and the second slide was kept in the laboratory as a back-up.

### 2.2. Ethics statement

Ethical clearance was obtained from the Institutional Ethics Committee of the Lambarene Medical Research Centre (CERMEL or *Centre de recherches médicales de Lambaréné*) in Gabon. The formal written approval was issued on 5 October 2022, under the reference number: CEI-020/2022.

### 2.3. Inclusivity in global research

Additional information regarding the ethical, cultural, and scientific considerations specific to inclusivity in global research is included in the Supporting Information (S18 Checklist).

### 2.4. Sample and study population

The study region is defined in terms of distance, in kilometers (km) from Lambaréné, the provincial capital of Moyen-Ogooué and covers villages to the north (up to 135 km from Lambaréné) and to the south (up to 75 km from Lambaréné). Within the study region, all villages were included in the study. The exact location of each village is depicted in panel A Fig 1 below. We aimed to survey the entire population between 18 and 65 years old of those villages. There was no required minimum number of participants per village. Given the high migratory patterns in the region, the village sample was not stratified by population and included communities with low and high prevalence of loiasis. The study planned to sample a minimum of 1,100 individuals in order to achieve a minimum detectable effect size of 6.5 percentage points at a 95 percent confidence interval (CI).

A

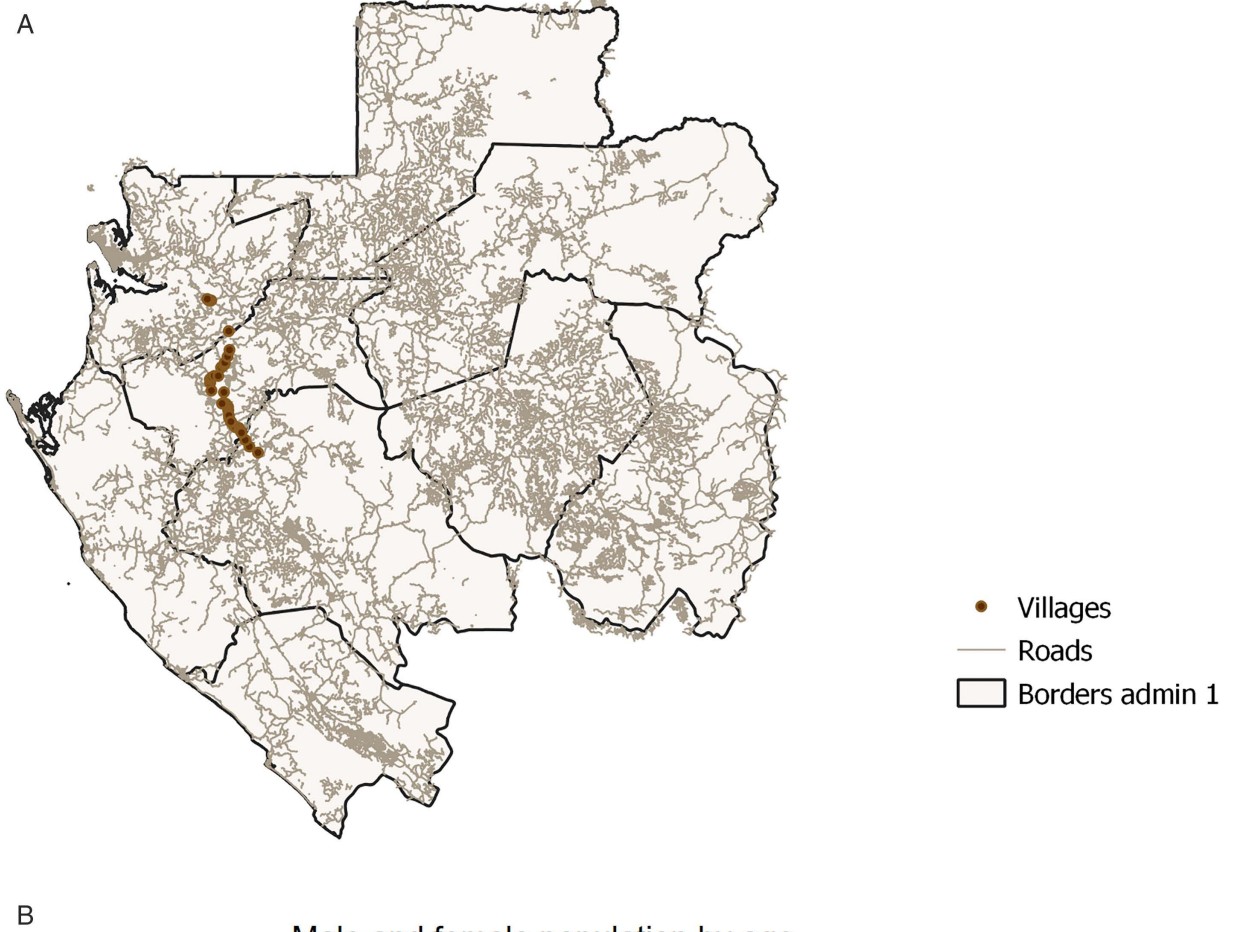

B

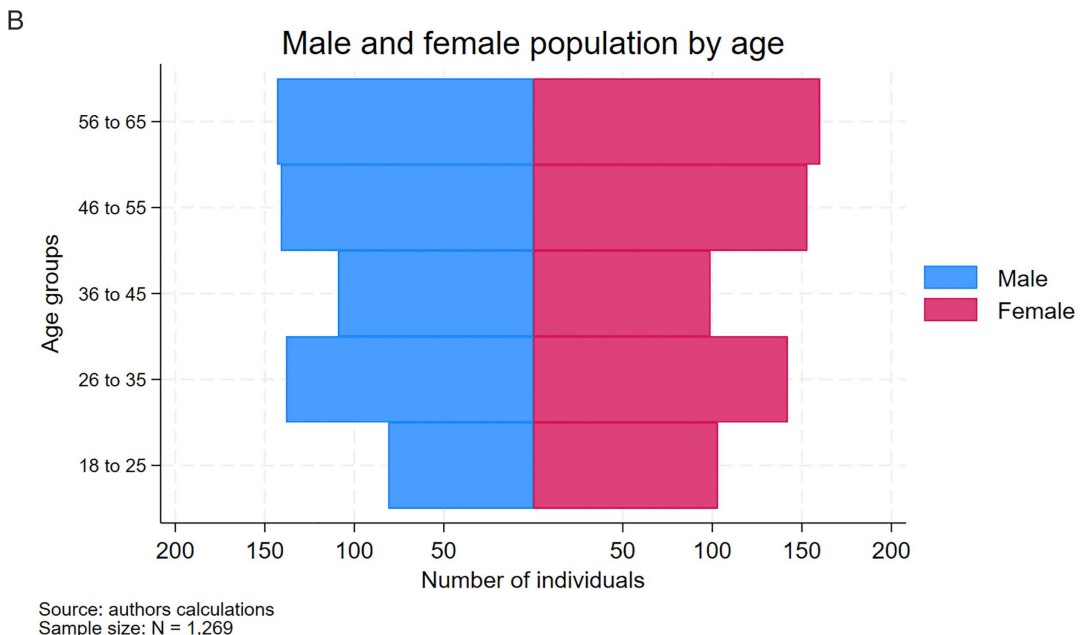

**Fig 1. Illustration of study sites and data.** Panel A: Location of surveyed villages in Gabon. Panel B: Age and demographic structure of sample population.

## 2.5. Loiasis status

In line with previous research, we constructed a binary indicator of loiasis status that takes the value 1 if a person is loiasis positive, either according to the biomarker (detectable microfilaremia in the thick blood smear) or the survey-based screening instrument "rapid assessment of prevalence of loiasis (RAPLOA)" [4,14]. The latter is a validated survey tool that assesses infection with adult worms allowing to correctly classify a person as loiasis positive in case of an occult infection [22]. RAPLOA itself consists of three questions, focuses on the self-reported migration of the adult worm through the conjunctiva of the eye, and involves showing the study subjects a typical picture of the eye worm migration. We classified study participants as loiasis positive according to RAPLOA if they reported to have been exposed to eye worm migration at least once during the 12 months prior to our survey.

## 2.6. Outcome variables

The construction of core variables is grounded in a Human Capital Approach (HCA). Estimates derived from HCA are commonly employed and imply that the economic burden of a disease takes into consideration the patient's perspective, meaning the whole period of loss of economic activity due to the disease [15,23]. In the following paragraphs, we flesh out the construction of key variables used in this cost-of-illness study with extensive details provided in S1 Table.

In total, we derived three major cost categories (direct medical costs, direct non-medical costs, indirect costs) which we ultimately added up to obtain an individual-level measure of a person's total health-related costs [15]. Specifically, direct medical costs consist of expenses related to inpatient and outpatient care, including but not limited to consultations by health professionals and traditional healers, pharmaceuticals and medical devices, or diagnostics. In contrast, direct non-medical costs represent health expenditures associated with the use of inpatient and outpatient care, but which do not directly relate to healthcare services (e.g., transportation costs, travel time, income loss for caregiver). Furthermore, we collected detailed information on how much participants spent for each of these categories at their last stay (in inpatient care) and/or last visit (outpatient care). Lastly, indirect costs refer to lost productivity due to (i) absenteeism and (ii) presenteeism. Absenteeism relates to the loss of income from not being able to work. During the survey, we asked participants about the number of days they could not work because of health problems. The reported days were used to estimate absenteeism. Presenteeism, on the other hand, captures income loss due to lower productivity at work because of loiasis. Our estimation of presenteeism relies on a short form of the Health and Labor Questionnaire (SF-HLQ) [24], which allows each respondent to assess their productivity, on a Likert scale from 0 to 10, during days in which they performed work although bothered by health problems (the detailed procedure for the calculation of days with reduced productivity is described in S14 Text). The days absent from work and those with reduced productivity were then multiplied by the daily wage of productive individuals in the region (S15 Text), for all concerned respondents, to estimate the monetary value of absenteeism and presenteeism, respectively.

## 2.7. Covariates

Our main specifications rely on demographic (age and gender), socio-economic (household wealth, completed education, work in the forest), health (a recent malaria infection), and spatial (village fixed effects) indicator(s) as controls. Details on the construction of the related variables are presented in S1 Table.

## 2.8. Statistical analysis

All cost estimates were derived by first applying entropy balancing in order to make observations in the control group (loiasis = 0) vs. treatment group (loiasis = 1) comparable in terms of observable characteristics. Weights are optimized to minimize differences in means (first moment order conditions). Then using the weights obtained from entropy balancing, we estimated weighted generalized linear models (GLM) to obtain estimates for the impact of loiasis on different cost

categories. This two-step process is considered the principal approach in deriving estimates in cost-of-illness studies [16,25–27]. Following common practice in the cost-of-illness literature and considering that the density function of various cost-items tends to be skewed [17,23,28], we estimated a GLM regression with a gamma distribution and a log link function [23,25,29,30]. To compute robust standard errors, we used the linearized variance estimator, also known as a first-order Taylor series linear approximation. Additional information on the statistical analysis can be found in S16 Text. All statistical analyses were performed using the software package STATA 18.5.

## 2.9. Sensitivity analysis

We performed several robustness checks: (i) assessing results depending on the underlying distribution function (gamma vs. negative binomial; S4 Table), (ii) optimization of weights based on alternative higher-order moment conditions (S5 Table), and (iii) different covariate specifications (S6 Table). Our results remain robust to these changes. Furthermore, in order to test for different roles of microfilariae versus adult worms, we vary the measure of loiasis, looking at (iv) RAP-LOA positive vs. RAPLOA negative (S7 Table), and (v) migratory vs. non-migratory loiasis (S8 Table). Migratory loiasis describes patients who exhibit clinical signs of adult worm movement—such as eye worm migration, Calabar swelling, or dermal migration— regardless of whether microfilaremia is confirmed. In contrast, non-migratory loiasis refers to cases where microfilaremia is detected without any signs of adult worm migration [12]. Finally, we re-assessed our main specification and controlled for the presence of chronic diseases (S9 Table). In order to assess potential spatial clusters related to loiasis infection and health costs, we map our main variables at the village level in S10 Fig. The visual inspection of the figure suggests that there are no obvious spatial correlations across villages. Therefore, we believe that our clustering approach is valid.

## 3. Results

In total, our analytical sample consists of 1,269 individuals from 38 villages who both successfully completed the survey and provided a blood sample. Descriptive results for the analytical sample are depicted in S2 Table. The average person in our sample is 43 years old, female (52 percent), completed at least secondary school (58 percent), lives together with 4 other household members, and is covered to some extent by public health insurance (78 percent). On average, per capita income is 68.73 USD per month, which is indicative of the economic challenges the population in the region faces. Furthermore, as shown in panel B of Fig 1, the gender and age-distribution in our analytical sample includes a relative equal number of observations in all gender-age brackets.

The prevalence of loiasis in our sample is high (S2 Table). About 47 percent of our study participants are infected with loiasis. We show in S3 Table that individuals tend to use inpatient and outpatient health services. Over a recall period of 3 months, we find that about 6 percent of individuals consulted an outpatient health service at least once, while 4 percent of individuals spent at least one day in inpatient care. On average, individuals incurred about 9.46 USD per month as out-of-pocket health-related expenses (which include active payment for health services and estimates of productivity loss). Furthermore, conditional on bearing any type of health-related costs (> 0 USD), i.e., direct or indirect costs, individuals bear a financial burden of about 35.53 USD per month, which is driven (among others) by absenteeism costs (22.60 USD) and costs related to consultation for inpatient care (18.91 USD).

We proceed by presenting results obtained from entropy balancing and subsequent GLM estimation on some intermediate outcome variables (Table 1 below). We find that, on average, loiasis did not lead to a significant increase in additional inpatient stays and outpatient visits. Nonetheless, we estimate that loiasis leads to about 3 additional days of work missed and more than 2 days of presenteeism per year. However, the related standard errors are large, and the estimates are not precisely estimated.

We find that loiasis leads to an increase in total costs of about 39.94 USD per year or about 5 percent of the average yearly per capita income (Table 2). As shown in the respective cost categories, the results are driven by higher

**Table 1. Impact of loiasis on absenteeism and presenteeism.**

| Variable | Observations | Estimates |
|---|---|---|
| | (1) | (2) |
| *Yearly estimates* | | |
| Days not worked | 1269 | 2.88  (3.00) |
| Days of presenteeism | 1269 | 2.31  (1.98) |
| Number stays inpatient | 1269 | 0.03  (0.07) |
| Number visits outpatient | 1269 | 0.03  (0.10) |

Notes: Variables are estimated over a 1-year period. Estimates refer to marginal effects and are obtained from a two-step process involving entropy balancing (step 1) and GLM (step 2). GLM refers to Generalized Linear Models and is specified with a negative binomial distribution and a log link function. Robust standard errors were used and are depicted in parentheses. */**/*** denote significance levels at 10/5/1 percent respectively.

**Table 2. Impact of loiasis on health costs.**

| Summary statistics | | | | Generalized Linear Model | | |
|---|---|---|---|---|---|---|
| Variable | Mean | Median | SD | Marginal effect | P-value | Confidence Interval |
| | (1) | (2) | (3) | (4) | (5) | (6) |
| **Total costs** | **163.64** | **0.00** | **924.46** | **39.94** | | |
| **Direct medical costs** | **92.80** | **0.00** | **822.45** | **9.17** | **0.750** | **−47.79–66.13** |
| *Inpatient care* | | | | | | |
| Consultation | 32.10 | 0.00 | 489.52 | | | |
| Medicine and devices | 11.16 | 0.00 | 100.45 | | | |
| Diagnostic | 5.48 | 0.00 | 82.27 | | | |
| Accommodation | 9.81 | 0.00 | 147.64 | | | |
| *Outpatient care* | | | | | | |
| Consultation | 13.93 | 0.00 | 286.61 | | | |
| Medicine and devices | 13.23 | 0.00 | 287.50 | | | |
| Diagnostic | 2.46 | 0.00 | 32.89 | | | |
| Additional medication | 4.64 | 0.00 | 78.22 | | | |
| **Direct non-medical costs** | **13.32** | **0.00** | **109.20** | **10.59** | **0.001***** | **2.16–19.01** |
| Informal payments (inp.) | 3.43 | 0.00 | 94.39 | | | |
| Informal payments (outp.) | 0.37 | 0.00 | 6.02 | | | |
| Transport (inp.) | 1.50 | 0.00 | 11.28 | | | |
| Transport (outp.) | 1.62 | 0.00 | 12.54 | | | |
| Time spent | 0.19 | 0.00 | 1.80 | | | |
| Caregiver | 6.22 | 0.00 | 36.01 | | | |
| **Indirect costs** | **57.52** | **0.00** | **170.89** | **20.18** | **0.098*** | **−4.71–45.08** |
| Absenteeism | 46.15 | 0.00 | 152.59 | | | |
| Presenteeism | 11.37 | 0.00 | 57.88 | | | |

Notes: The sample consists of 1,269 observations. Estimates refer to marginal effects and are obtained from a two-step process involving entropy balancing (step 1) and GLM (step 2). GLM refers to Generalized Linear Models and is specified with a gamma distribution and a log link function. 'Inp.' stands for inpatient care, while 'Outp.' refers to outpatient care. All expenditure values are in US Dollars. Robust standard errors were used. */**/*** denote significance levels at 10/5/1 percent respectively.

expenses in indirect costs (20.18 USD [95 percent CI = − 4.71–45.08] per individual per year) and direct non-medical costs (10.59 USD [95 percent CI = 2.16–19.01] per individual per year). Within the indirect costs, absenteeism represents the most important share of indirect costs, compared to presenteeism: the estimated absenteeism costs are 18.21 USD (p-value < 0.10). As for direct non-medical costs, they are mainly driven by the costs for caregivers (7.00 USD, p-value < 0.01). In the related prespecified sensitivity analyses, we found that our main results tend to hold irrespective of the particular changes in the model specifications (S4–S9 Tables). Comparing RAPLOA positive to RAPLOA negative individuals, regardless of their microfilaremia status (S7 Table), suggests that in particular the manifestation of adult worms and very apparent symptoms (as measured through the RAPLOA screening instrument) seems to be related to incurred costs. A similar insight is gained when comparing migratory vs. non-migratory loiasis, as suggested in S8 Table.

Next, we extrapolate our study results to derive a monetary estimate of the total annual, incremental costs of loiasis for the whole of Gabon. For this, we take into consideration regional differences in terms of population size and loiasis prevalence rates. The extrapolations accounts for the uncertainty in parameter input estimates by using bootstrap procedures (N = 5,000). Uncertainty is modelled for three parameters, namely the rural and urban prevalence rate of loiasis, and the rural-urban price ratio; a parameter we use to translate prices from our rural context to urban settings. We find that the annual, incremental costs of loiasis amount to 10,237,164.90 USD (sensitivity range = 8,657,408.5–11,822,899.00) for the entire country; considering a population of 2,484,789 inhabitants in the year 2023 (2,261,879 people in urban and 222,910 people in rural areas), a loiasis base prevalence rate of 5% in urban and 47% in rural areas, and a base urban-rural price ratio of 1.34 [4,31].

In Table 3, we re-estimate our main specification for different socio-demographic subgroups such as different age brackets and gender. Concerning age groups, we find that the impact of loiasis on health-related costs is particularly driven by younger individuals in the 18–25 and the 26–35 age groups. For the latter group, we estimate an incremental cost of 104.87 USD (p-value < 0.05) per individual per year in terms of indirect costs and 5.43 USD (p-value < 0.10) per individual per year in terms of direct non-medical costs. A similar trend is observed among the individuals aged from 18 to 25; in this group, the indirect costs amount to 38.37 USD (p-value < 0.05) per year and the direct non-medical costs are estimated to be 5.48 (p-value < 0.10) USD per year per individual.

With respect to gender differences, we find that loiasis-related costs are somewhat higher for women compared to men. We estimate the yearly indirect costs to be 72.58 USD (p-value < 0.01) and the direct non-medical costs to amount to 3.72 USD (p-value > 0.10) for women; whereas men affected by loiasis do not seem to incur significant indirect costs, but an estimated amount of 25.91 USD (p-value < 0.01) per year per individual in terms of direct non-medical costs. Additionally, splitting our sample at the median of household wealth shows that wealthier individuals incur higher loiasis related costs in total – driven by higher non-medical and indirect costs – whereas poorer individuals incur more direct medical costs (S11 Table). Non-working individuals bear higher indirect costs of up to 53.21 USD (p-value < 0.05) compared to non-working individuals (S12 Table).

**Table 3. Split-sample estimates on the impact of loiasis on health costs.**

| Variable | 18–25 | 26–35 | 36–65 | Women | Men |
|---|---|---|---|---|---|
| | (1) | (2) | (3) | (4) | (5) |
| Direct medical costs | −29.13 (39.54) | −24.04 (21.87) | 25.98 (32.07) | −36.03 (23.88) | 11.95 (21.70) |
| Direct non-medical costs | 5.48 (3.90)* | 5.43 (3.62)* | 8.11 (4.95)* | 3.72 (3.29) | 25.91 (15.18)*** |
| Indirect costs | 38.37 (21.35)** | 104.87 (75.63)** | −9.82 (12.39) | 72.58 (27.46)*** | −3.86 (15.09) |
| Observations | 184 | 280 | 805 | 657 | 612 |

Notes: Estimates refer to marginal effects and are obtained from a two-step process involving entropy balancing (step 1) and GLM (step 2). GLM refers to Generalized Linear Models and is specified with a gamma distribution and a log link function. All expenditure values are in US dollars. Robust standard errors were used and are depicted in parentheses. */**/*** denote significance levels at 10/5/1 percent respectively.

Lastly, we present in Table 4 estimates of the incremental costs of loiasis if individuals actively sought medical care. Measure 1 relates to self-reported expenditures on loiasis treatment, conditional on having used formal or informal health services. In total, 27 individuals reported to have actively looked for loiasis treatment at some point in the past. On average, these persons spent about 29.35 [Median = 20.07] USD per treatment (based on self-reports), with costs being driven by the medicines and medical devices, as well as the consultation and diagnostic sub-components.

Measure 2 is an unincentivized willingness-to-pay (WTP) measure in which individuals were asked how much they would spend on a medical treatment if they were infected with loiasis. Out of the 483 study participants who provided information, on average, the respondents reported that they were willing to spend about 29.95 [Median = 16.73] USD per treatment.

## 4. Discussion

Supported by recent evidence on the potentially substantial adverse health effects of loiasis on cognition, mortality, and general disease burden, several scholars have advocated for loiasis to be recognized as a NTD by the WHO [1,8–10,32]. While many of the health effects of loiasis are not yet well understood, even less is known regarding the financial burden and costs associated with the infection. Pursuing a cost-of-illness study, we present the first empirical evidence on the full individual-level costs of loiasis from a societal perspective.

We show that loiasis results on average in additional costs of 39.94 USD per year per individual. These are primarily driven by two major cost components, namely direct non-medical costs (10.59 USD) and indirect costs (20.18 USD), and by three major socio-demographic characteristics: women for indirect costs, men for direct non-medical costs, and people aged between 18 and 35 years. The latter finding can be rationalized by the circumstance that loiasis symptoms are particularly pronounced in early years of infection as people tend to a certain degree to develop immunotolerance later in life, while others might simply adopt and get used to live with the symptoms. In contrast, we do not find that loiasis affects other direct costs.

In our view, the obtained results help reconcile different perspectives on the health consequences of the illness. Bearing in mind that most of the loiasis patients are asymptomatic or suffer from painful and uncomfortable but somewhat manageable symptoms [1,33], health-related expenditures for the average loiasis patient are substantial though on average not necessarily catastrophic; a finding in line with some earlier views that loiasis' health impact is limited.

In contrast, and considering the more nascent literature on the potentially severe health impacts of loiasis [1,5,6,34], we find that in the case of loiasis-related acute symptoms and sufferings, seeking healthcare services is prohibitively expensive. In our study, self-reported costs for treatment amount on average to 29.35 USD per treatment if treated by a

**Table 4. Single-treatment loiasis healthcare costs: A patient's perspective.**

| Variable | Mean | Median | SD | Min | Max | Obs |
|---|---|---|---|---|---|---|
| | (1) | (2) | (3) | (4) | (5) | (6) |
| *Panel A: Self-reported expenditures for active loiasis treatment* | | | | | | |
| Total costs | 29.35 | 20.07 | 32.54 | 0.33 | 121.29 | 27 |
| Treatment, diagnostic, consultation | 19.27 | 16.73 | 16.92 | 1.67 | 58.55 | 22 |
| Drugs and devices | 22.69 | 16.73 | 16.68 | 1.67 | 58.55 | 13 |
| Accommodation and transport | 4.91 | 4.18 | 3.19 | 0.33 | 13.38 | 15 |
| *Panel B: WTP for loiasis treatment* | | | | | | |
| Willingness to Pay | 29.95 | 16.73 | 35.50 | 0.84 | 501.87 | 483 |

Notes: The table reports descriptive summary statistics. Panel A reports self-reported cost estimates from people who actively looked for loiasis treatment (N = 27). Panel B reports WTP estimates. WTP refers to non-incentivized measure of respondents' willingness to pay for loiasis. "SD" refers to standard deviation, "Obs" refers to the number of observations.

formal or informal medical provider, while individuals state on average a WTP of 29.95 USD per treatment (average WTP of the persons who reported past costs for loiasis treatment = 29.28 USD). The reported costs and the WTP agree to a large extent, therefore further substantiating the observation of the potential for high costs for symptomatic individuals. These numbers are compatible with and in range of a related study from rural Gabon that estimated the treatment costs for loiasis to amount to 58 USD per year per individual [14]. In summary, the costs for poor, rural individuals in Gabon to actively seek treatment for loiasis can be considered catastrophic, as they represent about 43 percent of monthly per capita income in the region (544 percent of monthly per capita income among the poorest 20% households) [35–37].

The finding that from an individual's perspective loiasis results on average in moderate, but in rare cases very high, additional costs allows to adopt more tailored health policy solutions. In particular, it strongly appeals to the rationale of health insurances in which risks for rare but extremely costly events are pooled among a larger number of individuals. Therefore, ensuring that individuals are covered by health insurance and that health insurance providers in Western and Central Africa cover diagnostics and treatment for loiasis seems important. Given the moderate average costs of loiasis, we believe that the inclusion of loiasis in health-insurance packages is financially viable for private, public, and community-based insurance systems.

For income-poor individuals to regularly spend an additional 39.94 USD per year is a relevant cost that can contribute to long-term, intergenerational poverty, as even moderate financial commitments have been found to lead to more risk aversion and less profitable agricultural investment strategies; hence facilitating a vicious cycle of poverty, less optimal investment in agriculture and healthcare, and ultimately worse downstream health outcomes and poverty status [38–40]. Consequently, each additional USD can result in substantially higher income losses over time. As such, our findings relate to the interconnections between NTDs and the United Nations' Sustainable Development Goals (SDGs), which have garnered increasing attention in recent years [41,42]. The debilitating symptoms associated with loiasis have been shown to compromise the achievement of several key SDGs, including SDG 1 (eradicating poverty in all its forms), SDG 2 (ending hunger and ensuring food security), SDG 3 (ensuring healthy lives and promoting well-being), and SDG 4 (ensuring inclusive and equitable quality education). Specifically, the disease's impact on individuals can lead to reduced productivity, limiting their ability to engage in agricultural activities, pursue employment, or attend school, thereby hindering progress towards these development goals.

With respect to our main results, we would like to point out that these are likely to represent lower-bound estimates of the actual cost of loiasis. The reason for this is threefold. First, the classification of loiasis positive and negative individuals is notoriously complex. Since no diagnostic gold standard is available with both serology- and PCR-based tests tending to misclassify individuals as loiasis negative or positive, a certain degree of measurement error was unavoidable [4,43,44]. Though we adopted best-practice classifications that do not exclusively rely on microfilariae detection in the blood but that take additionally into account symptom screening – screening for the migration of the worm along the eye (RAPLOA) –, it is possible that several loiasis positive individuals were not detected. As these individuals would now be included in the control group but incur loiasis related healthcare costs, this misclassification error unduly increases costs in the control group and could lead us to derive a lower-bound estimate on the true cost-of-illness for loiasis.

Second, loiasis is an infection that occurs in the rather very poor and less developed regions in each of the affected countries. In these regions, those involved in any kind of forest activities face a higher exposure to deerflies and are more prone to infection. In a setting in which a substantial share of the population needs to incur high transportation costs, and in which formal and informal healthcare services (e.g., clinics, hospitals, traditional healers) are expensive, we expect to observe an underutilization of formal and informal healthcare services. *Ceteris paribus*, being infected with loiasis would lead to higher costs if underutilization of health services was not present and if appropriate healthcare would be available in the healthcare system. Third, our estimate does not model explicitly the deterioration in overall health due to loiasis over a longer time horizon. For instance, previous studies have linked loiasis to a number of severe health complications such as organ failure, impaired cognition or mortality. In this regard our study suffer from selection bias in the sense that

individuals with impaired cognition were not admitted to our study, while individuals who died could not be included too. Assuming that these omissions present cases with comparatively higher loiasis-associated healthcare costs our main results can be seen as downward biased estimates.

Lastly, we would like to acknowledge and discuss further limitations of our study. First, cost-of-illness studies aim to derive causal estimates of a certain illness or health condition on costs. Given potentially strong ethical concerns and the high costs and logistical challenges of human challenge studies (controlled human infection models), our estimates are not based on a randomized controlled trial. Instead, we rely on quasi-experimental techniques that generate a synthetic control group conditional on observables (entropy-balancing in combination with GLM), which is usually superior to alternative matching approaches such as propensity score matching. Nevertheless, despite the balancing procedure and the inclusion of a broad set of covariates, residual endogeneity from unobserved factors cannot be fully ruled out. While our approach addresses issues such as forest workers' higher infection risk and differing care-seeking behaviour, it may not capture within-group heterogeneity—e.g., that more frequent forest users face higher risks and differ socioeconomically from those who go less often. Second, our indirect cost estimates are limited to absenteeism and presenteeism. This is because there appears to be no valid data on long-term disability and/or mortality due to loiasis. Given the very short periods of absenteeism (on average about 14 days per year in our sample for loiasis positive respondents), the choice of human capital approach or friction cost (and defining a friction period) did not matter. Third, the cross-sectional survey we implemented collects retrospective data on healthcare utilization, health expenditures, wages, and incomes. Since individuals tend to forget or misremember activities and payments, some additional measurement in core variables is likely to exist. In particular, since many forms of payments are informal and non-monetary, such payments might have been misremembered by respondents. However, this measurement error concerns both those individuals with loiasis and those without, which in general increases noise in the data and therefore makes it more difficult to obtain statistically significant estimates. Moreover, given that the study area involves low infrastructure development and low degrees of formalization, we believe that obtaining timestamp administrative data on formal and informal healthcare utilization or work transactions and earnings is highly infeasible. Fourth, there are several other factors that are likely to lead to higher standard errors and larger CIs in our cost estimates. In order to encourage participation in our study, we abstained from obtaining a fuller medical examination including additional testing. Instead, we collected various self-reported health indicators including a variety of infectious and chronic diseases. While our adopted approach is very common, we admit that additional measurement exists. Relatedly, a common concern in cost-of-illness studies is the presence of co-morbidities. If not adequately accounted for, the presence of co-morbidities can both upward and downward bias cost estimates. In our analysis we are able to conduct sensitivity checks around important co-morbidities, namely malaria and chronic illnesses and find that our main results continue to hold. Though the inclusion of co-morbidity variables affects parameter estimates, our main conclusion that loiasis leads to substantial illness-related costs remains. Despite the fact that we are not able to conduct sensitivity analyses for other forms of co-morbidities due to data constraints, we believe that the implemented checks suggest that loiasis leads to sizeable direct annual costs.

Regarding measurement concerns, the study area – like many parts of rural sub-Saharan Africa – is characterized by regular migration movements related to a number of factors such as seasonal work migration, marriages, and agricultural plots that involve longer travels. As such, concepts such as 'the local population', 'representative sampling' and 'selective migratory behaviour' (e.g., due to illness) are notoriously fuzzy and difficult to quantify. Fifth, while Gabon is geographically one of the largest loiasis endemic areas, globally it accounts for less than 2% of the total population at risk of loiasis infections. In this context, limitations for the generalizability of our study findings exist. Besides possible specifics of the selected study area within Gabon, the situation in other countries can differ substantially with respect to (i) healthcare supply and demand, and (ii) intensity of disease transmission. Extrapolation to other study settings should therefore be done with caution.

## 5. Conclusion

Our findings lend credence to the notion that loiasis should no longer be regarded as a benign condition with little impact on human health, and instead, it should be recognized as a significant public health concern for affected populations. We show that the average annual costs directly attributable to loiasis amount to 39.94 USD per individual per year. Average cost estimates are primarily driven by indirect costs and direct non-medical costs. We further show that in the rarer cases that individuals seek treatment at formal or informal healthcare providers for loiasis-specific symptoms, costs from the patient's perspective can be excessively high and amount to about 43 percent of the average monthly per capita income in the study region.

## Supporting information

**S1 Table. Construction and definition of core variables.**
(DOCX)

**S2 Table. Summary statistics of core variables.**
(DOCX)

**S3 Table. Summary statistics on the utilization of healthcare services.**
(DOCX)

**S4 Table. Alternative GLM estimates: Comparing gamma vs. negative binomial distribution.**
(DOCX)

**S5 Table. Alternative GLM estimates: Reweighting by different moment conditions.**
(DOCX)

**S6 Table. Alternative GLM estimates: Covariate specifications.**
(DOCX)

**S7 Table. RAPLOA positive vs. RAPLOA negative.**
(DOCX)

**S8 Table. Migratory vs. non-migratory loiasis.**
(DOCX)

**S9 Table. Impact of loiasis on health costs (controlling for chronic diseases).**
(DOCX)

**S10 Figure. Share of loiasis positive individuals and average total costs per village.**
(TIF)

**S11 Table. Impact of loiasis on health costs (by wealth group).**
(DOCX)

**S12 Table. Impact of loiasis on health costs (by working status).**
(DOCX)

**S13 Text. Adjustment related to health insurance.**
(DOCX)

**S14 Text. Estimation of presenteeism.**
(DOCX)

**S15 Text. Estimation of average daily wage.**
(DOCX)

**S16 Text. Entropy balancing and generalized linear model approach.**
(DOCX)

**S17 Text. Study instrument.**
(PDF)

**S18 Checklist. Inclusivity in global research.**
(DOCX)

## Acknowledgments

We would like to thank Jeanne Vallette d'Osia and Landry Yannick Nzengue for their outstanding research assistance. We are also thankful to the field team and coordination team at CERMEL that actively participated in the implementation of the study. Furthermore, we would like to express our gratitude to Isa Steiner for her inputs in the statistical analyses and to Carolin Brinkmann, Pia Michelitsch and Bertrand Lell for their feedback on previous draft versions of the paper.

## Author contributions

**Conceptualization:** Kerstin Perlik, Tom Stargardt, Olouyomi Scherif Adegnika, Michael Ramharter, Jan Priebe.

**Data curation:** Cédric Isaac Mbavu.

**Formal analysis:** Cédric Isaac Mbavu.

**Methodology:** Selidji Todagbe Agnandji, Olouyomi Scherif Adegnika.

**Resources:** Michael Ramharter.

**Software:** Cédric Isaac Mbavu.

**Supervision:** Tom Stargardt, Jan Priebe.

**Validation:** Tom Stargardt, Selidji Todagbe Agnandji, Rella Zoleko Manego.

**Writing – original draft:** Cédric Isaac Mbavu, Jan Priebe.

**Writing – review & editing:** Kerstin Perlik, Tom Stargardt, Selidji Todagbe Agnandji, Olouyomi Scherif Adegnika, Rella Zoleko Manego, Michael Ramharter.

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
