## [Decision Letter · Decision Letter 0]

23 Oct 2025

Dear Dr. Priebe,

**line numbering and a point-by-point response**  to the reviewers indicating which changes you made and on which lines.

We look forward to receiving your revised manuscript.

Kind regards,

David T. Zhu

Academic Editor

PLOS ONE

Journal Requirements:

7. We note that Figure 1 in your submission contain map/satellite images which may be copyrighted. All PLOS content is published under the Creative Commons Attribution License (CC BY 4.0), which means that the manuscript, images, and Supporting Information files will be freely available online, and any third party is permitted to access, download, copy, distribute, and use these materials in any way, even commercially, with proper attribution. For these reasons, we cannot publish previously copyrighted maps or satellite images created using proprietary data, such as Google software (Google Maps, Street View, and Earth). For more information, see our copyright guidelines: http://journals.plos.org/plosone/s/licenses-and-copyright.

Reviewers' comments:

Reviewer's Responses to Questions

**Comments to the Author**

1. Is the manuscript technically sound, and do the data support the conclusions?

Reviewer #1: Yes

Reviewer #2: Yes

2. Has the statistical analysis been performed appropriately and rigorously?

Reviewer #1: Yes

Reviewer #2: Yes

3. Have the authors made all data underlying the findings in their manuscript fully available?

Reviewer #1: Yes

Reviewer #2: Yes

4. Is the manuscript presented in an intelligible fashion and written in standard English?

Reviewer #1: Yes

Reviewer #2: Yes

Reviewer #1: The article by Mbavu et al. reports a study on the economic impact of loiasis in Gabon. This is likely the first study addressing this question, and although the approach is somewhat partial, it merits publication. There are no line numbers, which makes referencing specific comments difficult. Some references are unclear or not entirely appropriate—for example, references 12–15 for Gabon (Takougang and Pinder seem not well suited). This occurs in several parts of the manuscript. Similarly, the introduction lacks precision: it is unclear which references support the statement about USD 58. Shorter, clearer sentences and precise citations would help readers unfamiliar with loiasis to follow the logic and evidence. The manuscript also lacks the ethical approval number and any administrative authorization.

Regarding sampling, beyond the total target of 1,100 individuals, it is not clear if there was a minimum number of participants per village or a minimum proportion required. Concerning loiasis status, some results suggest different roles for microfilariae versus adult worms. Perhaps it would have been useful to include a sensitivity analysis using the same methodology applied to only mf+ vs mf- and RAPLOA+/mf- vs RAPLOA-/mf-. Adding this or discussing it in the discussion could provide additional insight.

From a methodological perspective, I am not an expert in health economics, but several points are worth noting. The study uses a solid approach with entropy balancing and generalized linear models (GLM), which is generally accepted in cost-of-illness studies. The distinction between direct medical, direct non-medical, and indirect costs is clear. Including both formal and informal healthcare expenditures, such as traditional healers, is particularly valuable given the local context. Using national health insurance data (CNAMGS) to validate self-reported expenditures and converting to USD adds transparency.

However, there are some limitations. The cross-sectional design only captures short-term costs and cannot assess causality. Chronic or secondary complications, such as renal problems, neurological effects, or other organ dysfunctions, are not included, which may underestimate the total societal burden. It would be helpful for the authors to clarify whether the goal was to measure short-term individual costs or broader societal costs, and especially to discuss how long-term complications might change the estimates.

The indirect cost estimates rely on self-reported data on days missed or reduced productivity, which may not be precise. It is also unclear how the authors assigned a monetary value to the work of participants without a fixed salary, such as subsistence farmers. Additionally, it would help to clarify whether these costs are calculated from the perspective of the individual, the household, or society as a whole.

Another concern is comorbidities. While malaria is included as a covariate, other common conditions like Mansonella perstans or chronic diseases may also affect both infection and healthcare costs. Clarifying whether these were considered would strengthen the analysis. Differences in cost distribution by gender or wealth could also be explored.

The authors tested the robustness of their GLM results by comparing different distributions (gamma vs. negative binomial) and alternative covariate specifications. This is reassuring, although it would be useful to briefly comment on whether these alternatives produced notably different results or just minor changes. Village fixed effects are included, but potential spatial correlation between villages is not discussed. How hospital and outpatient costs were constructed at the unit level could also be clarified: were expenditures broken down by consultations, diagnostics, medications, etc.?

Overall, the methodology is strong but could benefit from a clearer explanation of perspective, more details on productivity loss measurement, and discussion of cross-sectional limitations and excluded chronic complications.

Results and discussion are generally clear. The presentation of both aggregate and stratified analyses is helpful. However, the interpretation of USD 40 annual additional cost per person could be clarified: does it include only short-term costs, or could longer-term complications change the estimate? The higher costs observed among women and younger adults are interesting, but alternative explanations such as labor patterns or care-seeking behavior could also be discussed.

The discussion on including loiasis in insurance packages is relevant. Extrapolation to national costs is reasonable but should account for uncertainty; sensitivity ranges would help. Limitations are well acknowledged, but endogeneity between infection status and health costs could be discussed further.

In summary, this study provides valuable empirical evidence on the economic burden of loiasis using appropriate methods for a cost-of-illness study and deserves to be published. Its main limitations are the short-term focus, reliance on self-reported data, and exclusion of severe complications. Including discussion or estimates of these additional costs, and clarifying perspective and measurement details, would strengthen interpretation and policy relevance.

Reviewer #2: The manuscript is on an important topic for the disease of concern.

In general, the manuscript is well written and easy to follow.

However, I do have some minor comments:

1. The funding statement has to be consistent throughout the manuscript: Sometimes no funding, and sometimes there is funding. Please proceed to update the manuscript throughout.

2. In the introduction section, page #2, the following statement (According to the authors’ preferred specifications, loiasis resulted in treatment costs of 58 United States Dollar (USD) per person per year) needs a reference:

3. In the core manuscript, the ethical approval number is missing, while this is mentioned in the submission system.

4. Figure 1, Panel A The map can be included in the Country Gabon map, including the names of localities, if possible.

5: To better weigh the burden of the additional cost of Loaisis on health expenses, it will be interesting for the author to report on the average annual income of the population of interest, including any in-kind and cash income, if available.

**Do you want your identity to be public for this peer review?** For information about this choice, including consent withdrawal, please see our Privacy Policy

Reviewer #1: No

Reviewer #2: No

---

## [Author Response · Author response to Decision Letter 1]

23 Dec 2025

Re-submission of:

The economic burden of loiasis: a comprehensive cost-of-illness analysis of regionally representative, individual-level data from rural Gabon

Responses to the academic editor and reviewers

December 06, 2025

Dear academic editor Mr. David T. Zhu,

We thank you for the opportunity to revise and resubmit our manuscript to PLOS ONE. Moreover, we would like to express our gratitude to both reviewers and you for the time taken to review our manuscript and for the constructive comments everyone offered.

In our revised re-submission, we have sought to address each reviewer’s insightful comments. Ultimately, we believe that the revised version of the manuscript constitutes a stronger contribution to the field.

In what follows, we detail two main changes we implemented on the manuscript before providing a line-by-line response to each of the reviewers and you. Please note that we refer to reviewers in terms of the journal’s numbering: Reviewer #1 and Reviewer #2.

1. Additional analyses: both reviewers provided valuable input on how to further explore and underscore the robustness of our results. In response to these comments, we expanded our robustness checks in multiple ways. We briefly summarize these as follows:

a. Measurement of loiasis (I): RAPLOA positive vs. RAPLOA negative

b. Measurement of loiasis (II) Migratory vs. non-migratory loiasis

c. Alternative model specification: Controlling for respondents’ chronic diseases

d. Heterogeneous effects (I): Impact of loiasis on different socio-economic groups

e. Heterogenous effects (II) Impact of loiasis by work status

f. Sensitivity analysis around the uncertainty of parameter estimates used in the extrapolation of Gabon-wide costs

2. Clarifications and precision in the write-up: both reviewers pointed out a number of imprecisions and lacking explanations in our manuscript. We took all of these comments very seriously and aimed to address them. We believe that the revised manuscript is now more concise.

Responses to the academic editor

1. Thank you for submitting your manuscript to PLOS ONE. After careful consideration, we feel that it has merit but does not fully meet PLOS ONE’s publication criteria as it currently stands. Therefore, we invite you to submit a revised version of the manuscript that addresses the points raised during the review process. When you resubmit your manuscript, please include line numbering and a point-by-point response to the reviewers indicating which changes you made and on which lines.

We would like to thank the academic editor for giving us this opportunity. As requested, we have modified the manuscript by including line numbering. In addition, we have prepared a point-by-point response letter to the academic editor and reviewers.

2. We would like to keep the financial disclosure made available in the first version of the submitted manuscript, which states the following (lines 50 – 52):

“Funding to support this research was provided by the Bernhard Nocht Institute for Tropical Medicine (BNITM) and the German Center for Infection Research (FKZ: 8008803909).”

We thank you for sharing this information.

We will upload the laboratory protocol in due time. Please be assured that it is available.

5. Please ensure that your manuscript meets PLOS ONE’s style requirements, including those for file naming. The PLOS ONE style templates can be found at https://journals.plos.org/plosone/s/file?id=wjVg/PLOSOne_formatting_sample_main_body.pdf and https://journals.plos.org/plosone/s/file?id=ba62/PLOSOne_formatting_sample_title_authors_affiliations.pdf

We thank you for sharing this useful information. The manuscript has been adapted accordingly and hopefully now meets all PLOS ONE’s style requirements.

6. Please include a complete copy of PLOS’ questionnaire on inclusivity in global research in your revised manuscript. Our policy for research in this area aims to improve transparency in the reporting of research performed outside of researchers’ own country or community. The policy applies to researchers who have travelled to a different country to conduct research, research with Indigenous populations or their lands, and research on cultural artefacts. The questionnaire can also be requested at the journal’s discretion for any other submissions, even if these conditions are not met. Please find more information on the policy and a link to download a blank copy of the questionnaire here: https://journals.plos.org/plosone/s/best-practices-in-research-reporting. Please upload a completed version of your questionnaire as Supporting Information when you resubmit your manuscript.

We thank you for this suggestion. The questionnaire has been filled in and will be submitted along with the other documents.

7. We note that the grant information you provided in the ‘Funding Information’ and ‘Financial Disclosure’ sections do not match. When you resubmit, please ensure that you provide the correct grant numbers for the awards you received for your study in the ‘Funding Information’ section.

We would like to thank you and apologize for this inconsistency. We have adjusted the ‘Funding Information’ and ‘Financial Disclosure’ as necessary.

8. Thank you for stating the following financial disclosure: “The author(s) received no specific funding for this work.”

We would like to thank you for this advice. We included the requested information and clarification in the updated cover letter. Moreover, we provide this information here now as well:

Please find below a response to each of the listed points:

a) Funding to support this research was provided by the Bernhard Nocht Institute for Tropical Medicine (BNITM) and the German Center for Infection Research (FKZ: 8008803909).

c) The following authors received a salary from BNITM: Cédric Isaac Mbavu, Kerstin Perlik, Rella Zoleko-Manego, Michael Ramharter and Jan Priebe.

d) Not applicable.

9. When completing the data availability statement of the submission form, you indicated that you will make your data available on acceptance. We strongly recommend all authors decide on a data sharing plan before acceptance, as the process can be lengthy and hold up publication timelines. Please note that, though access restrictions are acceptable now, your entire data will need to be made freely accessible if your manuscript is accepted for publication. This policy applies to all data except where public deposition would breach compliance with the protocol approved by your research ethics board. If you are unable to adhere to our open data policy, please kindly revise your statement to explain your reasoning and we will seek the editor’s input on an exemption. Please be assured that, once you have provided your new statement, the assessment of your exemption will not hold up the peer review process.

Yes, we confirm that we will make our data available upon acceptance of the manuscript. Furthermore, we are familiar with posting such data and replication files on public repositories. Lastly, we already have finalized our data sharing plan but would be open to adjust the plan in case of special requirements by the journal.

10. Please include your full ethics statement in the ‘Methods’ section of your manuscript file. In your statement, please include the full name of the IRB or ethics committee who approved or waived your study, as well as whether or not you obtained informed written or verbal consent. If consent was waived for your study, please include this information in your statement as well.

Thank you for this advice. We acknowledge that the previous ethics statement was already present in the ‘Methods’ section but was not yet complete. In response, we added the additional content in the respective section paragraph.

11. We note that Figure 1 in your submission contain map/satellite images which may be copyrighted. All PLOS content is published under the Creative Commons Attribution License (CC BY 4.0), which means that the manuscript, images, and Supporting Information files will be freely available online, and any third party is permitted to access, download, copy, distribute, and use these materials in any way, even commercially, with proper attribution. For these reasons, we cannot publish previously copyrighted maps or satellite images created using proprietary data, such as Google software (Google Maps, Street View, and Earth). For more information, see our copyright guidelines: http://journals.plos.org/plosone/s/licenses-and-copyright.

We thank you for these comments and suggestions. Figure 1 (Panel A) plots the location of study villages on an administrative map of Gabon provided by GADM. GADM allows the use of their data to create maps and figures for publishing academic research. We did not use any licensed satellite images. Therefore, this content can be published under the Creative Commons Attribution License (CC BY 4.0).

12. Please include captions for your Supporting Information files at the end of your manuscript, and update any in-text citations to match accordingly. Please see our Supporting Information guidelines for more information: http://journals.plos.org/plosone/s/supporting-information.

We thank you for this comment. We included the captions at the end of the manuscript for the supporting information and updated the text accordingly.

None of the reviewers recommended to cite specific previously published works.

We thank you for this valuable contribution. We reviewed the references list and ensured that it is complete and correct. We made one formal change to the sources by adding the names “Alleyne GA, Cohen D” in the publication on “Health, economic growth, and poverty reduction” by the WHO (line 704).

Two references have been deleted:

• The former reference [21]: WHO. ESPEN 2019 Annual Report. World Health Organization, 2019.

• The former reference [33]: Group WB. World Development Indicators 2023 [cited 2025 05 March]. Available from: https://databank.worldbank.org/source/world-development-indicators.

Two other references have been added:

• Reference [12]: Ramharter M, Schlabe S, Hubner MP, Michelitsch P, Kurth F, Belard S, et al. Diagnosis, management and prevention of loiasis: guideline of the German Society for Tropical Medicine, Travel Medicine, and Global Health (DTG). Infection. 2025;53(3):851-72. Epub 20250521. doi: 10.1007/s15010-024-02443-2. PubMed PMID: 40397272; PubMed Central PMCID: PMCPMC12137371.

• Reference [20]: Kelly-Hope LA, Hemingway J, Taylor MJ, Molyneux DH. Increasing evidence of low lymphatic filariasis prevalence in high risk Loa loa areas in Central and West Africa: a literature review. Parasit Vectors. 2018;11(1):349. Epub 20180615. doi: 10.1186/s13071-018-2900-y. PubMed PMID: 29907117; PubMed Central PMCID: PMCPMC6004093.

Lastly, we did not cite papers that have been retracted.

Responses to Re

---

## [Editor Report · Decision Letter 1]

25 Dec 2025

The economic burden of loiasis: a comprehensive cost-of-illness analysis of regionally representative, individual-level data from rural Gabon

PONE-D-25-36336R1

Dear Dr. Priebe,

We’re pleased to inform you that your manuscript has been judged scientifically suitable for publication and will be formally accepted for publication once it meets all outstanding technical requirements.

Kind regards,

David T. Zhu

Academic Editor

PLOS One

Additional Editor Comments (optional):

The authors did an excellent job responding to Reviewer 1 and Reviewer 2's comments, providing a detailed response letter, clarifications to the methodology and limitations, and additional analyses based on reviewers' comments (e.g., case definition based on RAPLOA and migratory status). After considering these revisions and the updated manuscript, I am pleased to accept it for publication.
---

## [Editor Report · Acceptance letter]

PONE-D-25-36336R1

PLOS One

Dear Dr. Priebe,

I'm pleased to inform you that your manuscript has been deemed suitable for publication in PLOS One. Congratulations! Your manuscript is now being handed over to our production team.

Kind regards,

on behalf of

Mr. David T. Zhu

Academic Editor

PLOS One